# Direct Measurement of Heat Produced during Drilling for Implant Site Preparation

**Yongsoo Kim [1], Sungwon Ju [1], MinJu Kim [1]** **, Minsu Park [1], Sangho Jun [2] and Jinsoo Ahn [1,*]**

[1] Dental Research Institute and Department of Dental Biomaterials Science, School of Dentistry, Seoul National University, 101 Daehak-ro, Jongro-gu, Seoul 03080, Korea; kyst4@snu.ac.kr (Y.K.); swju27@snu.ac.kr (S.J.); kkotemandu@snu.ac.kr (M.K.); bimmer740@naver.com (M.P.)

[2] Department of Oral and Maxillofacial Surgery, Anam Hospital, Korea University, 73 Inchon-ro, Seongbuk-gu, Seoul 02841, Korea; junsang@korea.ac.kr

[*] Correspondence: ahnjin@snu.ac.kr; Tel.: +82-2-740-8691

**Featured Application: The result of this paper provides clinical guidance for the proper use of the drill during implant site preparation.**

**Abstract:** The aim of this study was to directly measure the temperature of the drill during implant site preparation. The measured temperature was compared to that previously reported inside the bone. The temperature change of the drill was measured using a thermocouple inserted inside the stainless steel drill using an 18-gauge needle and mercury-containing slip ring. Three thermocouples were inserted into the bone with different depths of 10 mm, 5 mm and 1 mm. The thermocouple was 0.5 mm away from the periphery of the drilled hole with a diameter of 3.4 mm. The drill rotating speed varied from 500 rpm to 2000 rpm. Each drilling procedure was performed 10 times, and the average was calculated. The temperature of the drill increased rapidly, and the thermocouples in the bone reached a maximum temperature after the drill temperature started to decrease. The maximum inner temperature of the bone was the highest at a depth of 10 mm. The patterns of the temperature change were similar at different rotating speeds. The actual maximum temperature at the drill and bone interface was significantly higher than the temperature measured inside the bone in previous reports.

**Keywords:** dental implant; site preparation; drilling heat

## 1. Introduction

Medical and dental implants involve placing a fixture in the space prepared by osteotomy. The space is then replenished by osseointegration, which is a direct functional and structural connection between the bone and the implant [1]. Drilling is undeniably the most essential process during implant site preparation, as success of osseointegration is determined at this stage. Mechanical and thermal damage to the bone is inevitable during drilling [2,3]. More than one minute of drilling at a temperature higher than 47 °C causes local osteonecrosis and impairs osseointegration [3,4]. The bone tissue is absorbed and degenerates into fat cells that are replaced by less differentiated cells [5,6]. These impaired bone cells lack the ability of proper osseous remodeling [7].

Previous studies have attempted to measure the temperature of the heat generated by the friction between the bone and the instrument during drilling in clinically relevant situations [8]. They used artificial bones and human cadaver bones to evaluate drilling methods with less excessive thermal damage [9,10]. However, most of the studies did not measure the heat directly from the interface. The measurements were taken with thermocouples inserted into the bone located >0.5 mm away from the drilling interface [11,12]. A few studies measured the heat from the drill instead of from the bone

to avoid potential errors associated with the different thermal conductivities between bones and the stainless steel of the drill [13–15]. However, the potential measurement errors associated with (1) the friction between the drill and thermosensor located in the middle of the drill and (2) the air thermal insulation of the gap between the thermocouple, were not considered in the previous works [14,15].

Therefore, a more clinically relevant thermal setup is essential to measure the actual heat generated by the implant drilling related to the drilling depth, speed, time, and pressure without the thermal insulation effect of bone and frictional heat between the drill and the thermosensor [16,17]. In this study, the heat of the drilling was measured by the rotating thermosensor that is directly connected to the drill. The temperature measured directly on the drill was compared with previous methods that measured the temperature in the bone and in the drill. In addition, the heat produced at the moment of drilling was evaluated as a function of drilling speeds and depths, and the surface temperature of the drill was monitored using infrared analysis. The null hypothesis of this study was that there is no significant difference in temperature among different drilling speeds and depths. This study provides more sophisticated and clinically relevant heat measurements for more successful implant procedures.

## 2. Materials and Methods

To simulate drilling in clinical implant situations, we custom-built our equipment and the experiments were conducted using an artificial bone model (saw bone #1522-27, Pacific Research Laboratories Inc., Vashon, WA, USA).

### 2.1. Themosensor Located in Bone for the Heat Evaluation

Based on previous methods, three thermocouples (TT-K-36 K-type, Omega, Stanford, USA) were inserted into each hole (1 mm diameter) placed 0.5 mm lateral to the perforations in the bone with different lengths of 1, 5, and 10 mm [18]. To avoid electronic interference between sensors, insulated silicone coating (Silicone Varnish, Spray, S-830 NABAKEM®, Sihung, Korea) was applied to the thermocouples. Thermoconductive paste (Super Lube®, New York, NY, USA) was applied to the thermocouples to prevent the error induced by heat loss or the outer environment, and the coated thermocouples were inserted into the holes. Heat produced from removal of the bone was measured and recorded by the thermocouple.

### 2.2. Themosensor Located in Contact with the Proximal End of the Drill for the Heat Evaluation

Based on previous methods, a thermocouple was inserted into the proximal end of the drill made of stainless steel (Zimmer Dental Inc., Carlsbad, CA, USA) using an 18-gauge needle and placed in contact with the distal drill to measure the temperature of the drill itself [14,15]. In this case, the heat generated from the friction between the thermosensor and the proximal end of the drill. The other end of the thermocouple was connected to the computer by a signal conditioner circuit (Thermocouple conditioner and Setpoint controller, Analog Devices, AD597, Norwood, MA, USA) and a data acquisition board (NI cDAQ-9174, NI 9205, National Instrument, Austin, USA). The thermocouple diameter was 0.019 inches, the accuracy was 0.4%, and the responding speed was 3 m/s.

### 2.3. Rotating Themosensor in Complete-Contact with the Drill for the Improved Heat Evaluation

Whereas the previous drill temperature measurement method described above ignored the heat generated from the friction between the thermocouple and the proximal end of the drill, we custom-built a rotating thermocouple with a sling ring at the end of the thermocouple that can rotate with the drill (Figure 1a) [14,15]. This new method successfully eliminated the heat from friction (which causes an average 5 °C increase; details are demonstrated in the Results section in this paper). A mercury-containing slip ring (A2S, AK-Slip Ring, Inchon, Korea) was also used to deliver the signal measured at the rotating drill to the processing unit without heat loss (Figure 1a). Another thermocouple was placed to measure the ambient temperature. Infrared ray sensors (DTS-150, DIWELL, Gunpo, Korea) were located around the drill hole to analyze the changes in the temperature

of the drill surface at the time of drilling; that is compliment to the direct temperature measurement using the rotating thermocouple inside the drill. All measured temperatures were recorded and saved on a personal computer using Labview 2009 software (National Instrument) (Figure 1b).

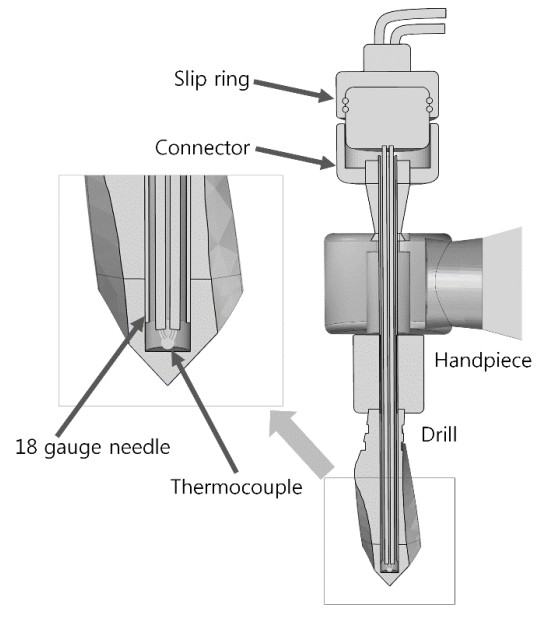

(**a**)

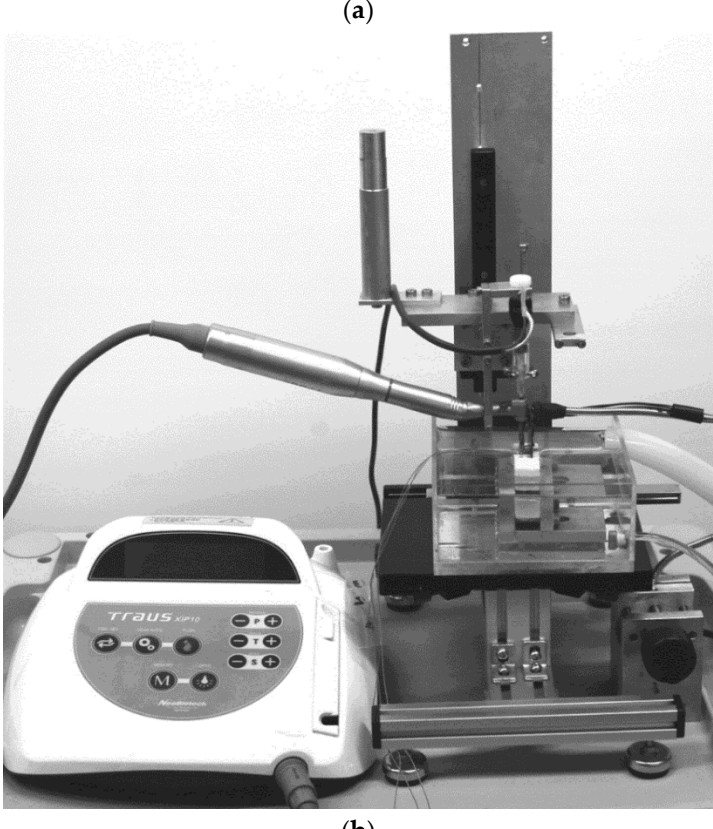

(**b**)

**Figure 1.** (**a**) A sectional view of the drill and slip ring assembly; (**b**) A custom-made experimental device and drilling motor.

### 2.4. Equipment Setup

A surgical handpiece (Strong acl-41I, Saeshin Precision Co., Ltd., Daegu, Korea) was fixed to a stationary base to allow vertical movement of the drill. An isotonic saline (0.90 *w/v* % of NaCl) bath was located below the surgical handpiece, and equipped with a lock device used to fix the bone block and restrict movement during drilling. Precise adjustment of the location of the bone block was feasible with the 3D adjusting table placed below the bath. A tube connected a thermostat-controlled water bath to maintain a consistent water level and temperature at 37 °C (Figure 1b). The inferior half of the bone installed with the thermocouple was submerged below the solution to simulate the clinical environment of the mouth. An infrared ray sensor was placed to measure the surface temperature simultaneously during the heat measurement with the thermocouples. The surgical handpiece was operated using the implant drilling motor (Traus xip10, Neobiotech, Seoul, Korea). The gear ratio was 20:1, insertion torque was at 30 N, and the drilling speed was 1200 rpm, which is commonly recommended by the manufacturers. An 800 g calibration weight was placed on top of the handpiece to maintain the consistent compressive force (Figure 1b). The drilling was immediately stopped once the specified depth was reached. Once the temperature became stable for 30 s, the thermosensors started to record the temperature. The same measurement was taken at different drilling speeds of 500 rpm and 2000 rpm to recreate clinical situations when slower and faster drilling is performed by clinicians. We did not use water cooling (irrigation) to avoid erroneous measurement of the temperature of water by the infrared ray sensor. The procedure was repeated 10 times, and the average and standard deviation were calculated.

### 2.5. Statistical Analysis

The maximum temperature value was statistically analyzed using one-way ANOVA. Scheffe's multiple comparison test was used to verify the significance between groups at $\alpha = 0.05$ (SPSS 20.0, IBM, Chicago, IL, USA).

### 3. Results

The average maximum temperature for each rotating speed and region are shown in Figure 2, and the temperature changes at 500, 1200 and 2000 rpm are shown in Figure 3. The direct drilling temperature without the rotating slip ring on the thermosensor, used in previous works, recorded an average 8 °C higher than our method with the rotating slip ring at all speeds. The mean values of peak temperature by drill speed and measuring depth is shown in Table 1.

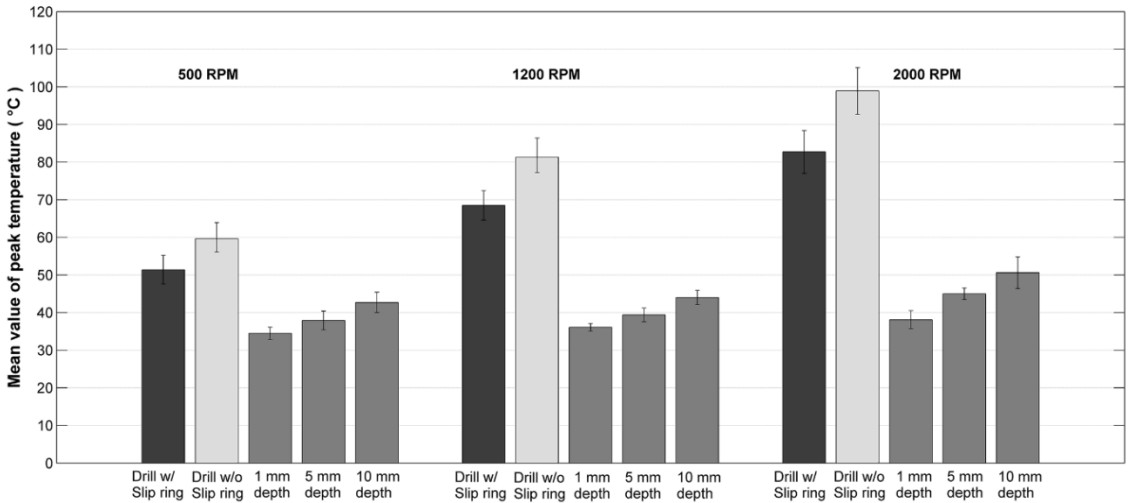

**Figure 2.** Comparison of implant drilling temperature measured at drill with/without slip ring or inside bone at a depth of 1, 5, and 10 mm, respectively, as a function of rotation speed.

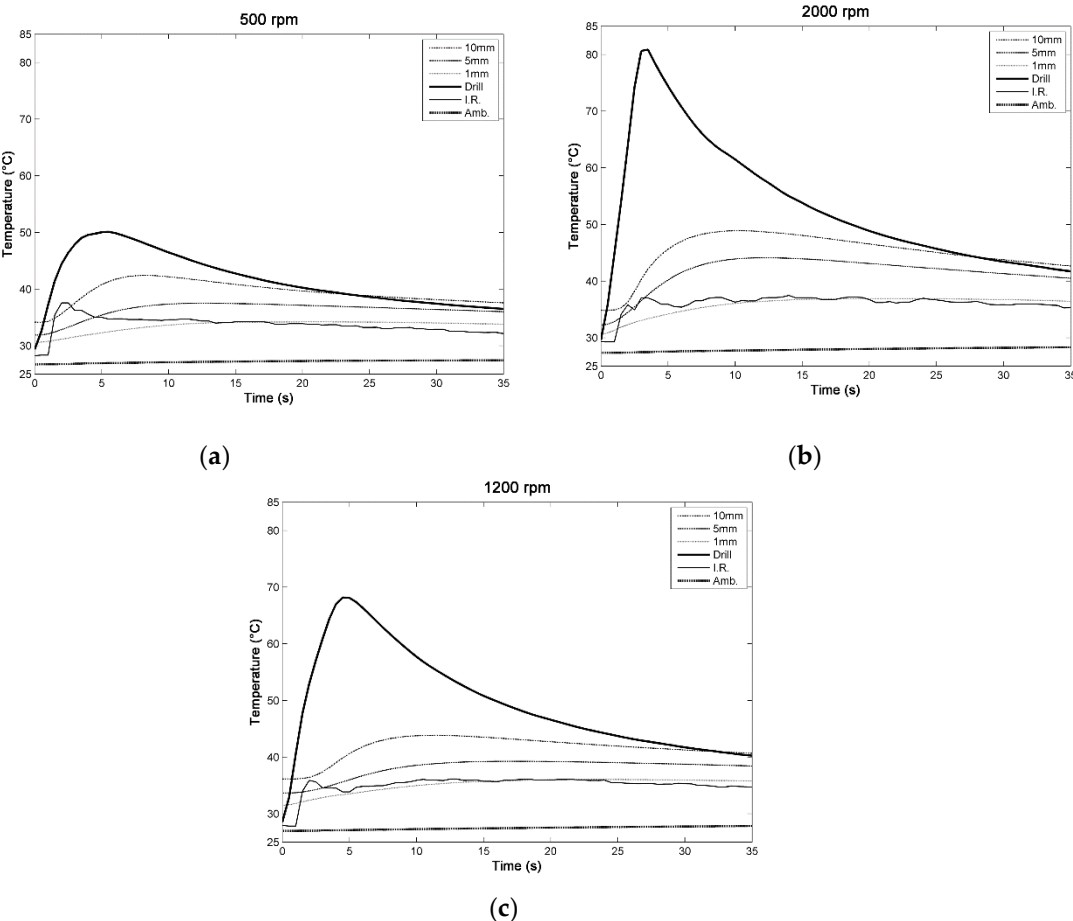

**Figure 3.** (**a**) The mean values of thermal change during drilling at 500 rpm; (**b**) at 1200 rpm; (**c**) at 2000 rpm.

**Table 1.** Mean values of peak temperature by drill speed and measuring depth.

| RPM | Drill | 1 mm Depth | 5 mm Depth | 10 mm Depth | IR |
|-----|-------|-----------|-----------|------------|-----|
| 500 | [a] 51.4 ± 3.8 | [a] 34.5 ± 1.6 | [a] 37.9 ± 2.5 | [a] 42.7 ± 2.7 | [a] 37.7 ± 2.9 |
| 1200 | [b] 68.5 ± 3.9 | [a] 36.1 ± 1.0 | [a] 39.4 ± 1.8 | [a] 44.0 ± 1.9 | [a] 38.1 ± 2.4 |
| 2000 | [c] 82.7 ± 5.7 | [b] 38.1 ± 2.4 | [b] 45.0 ± 1.5 | [b] 50.6 ± 4.2 | [a] 39.9 ± 3.4 |

[a,b,c] Different letters in each column indicate statistical significance among groups ($p < 0.05$).

When the drilling speed was 500 rpm, the temperature of the drill increased rapidly once the drilling started, and the mean maximum temperature of 51.4 °C, SD (standard deviation) 3.8, was reached in 5 s when the drill reached a depth of 10 mm. On the other hand, the thermocouples in the bone reached the maximum temperature (mean 42.7 °C, SD 2.7) after 7 s.

When the drilling speed was 1200 rpm, the temperature of the drill increased rapidly once the drilling started, and the maximum temperature (mean 68.5 °C, SD 3.9) was reached in 5 s when the drill reached a depth of 10 mm. On the other hand, the thermocouples in the bone reached the maximum temperature (mean 44.0 °C, SD 1.9) after 9 s.

When the drilling speed was 2000 rpm, the temperature of the drill increased rapidly once the drilling started, and the maximum temperature (mean 82.7 °C, SD 5.7) was reached in 4 s when the drill reached a depth of 10 mm. On the other hand, the thermocouples in the bone reached the maximum temperature (mean 50.6 °C, SD 4.2) after 9 s.

The patterns of temperature change at all speeds were very similar to one another. The maximum temperature measured inside the bone was at 10 mm depth compared to those at 5 mm and 1 mm depths.

The temperature differences at 1 mm depth as a function of drilling speed were very small (mean 34.5 °C, SD 1.6, to mean 38.1 °C, SD 2.4), whereas those at 5 mm depth (mean 37.9 °C, SD 2.5, to mean 45.0 °C, SD 1.5) and 10 mm (mean 42.7 °C, SD 2.7, to mean 50.6 °C, SD 4.2) depth were much more pronounced. The surface temperature measured using the infrared ray sensor was corroborated to the temperature measured at 1 mm depth in bone.

The maximum drill temperature increased rapidly within 4–5 s, whereas the temperature measured inside the bone reached the maximum temperature after 7–9 s regardless of the drill rotating speed. The thermocouple located in the bone indicated the maximum temperature at a depth of 10 mm when the drill temperature started decreasing, showing the disparity. The temperature measured on the drill decreased significantly after the drill stopped, whereas the temperature measured in bone decreased slowly during the first 5 s after the drill stopped.

We confirmed that the maximum values were statistically significantly different at each rotating speed, and the maximum temperature differed depending on the depth ($p < 0.05$).

No significant differences in maximum temperatures were observed between 500 rpm and 1200 rpm. However, the maximum temperature at 2000 rpm was significantly different from those at 500 rpm and 1200 rpm. The ambient temperature was not significantly changed and remained fairly consistent.

## 4. Discussion

In dental implant procedures, understanding and minimizing the heat generated by friction at the interface between the drill and bone based on a well-designed experiment is important to avoid the potential damage to native bone tissue because thermally damaged osseous tissues around the implant constrain osseointegration, and may lead to implant failures and revisions that are painful to both patients and clinicians [3,4,7,9,19]. Many previous studies have tried to evaluate the excessive heat during drilling bones, but the methods to measure the temperature changes are somewhat irrelevant to clinical situations. For examples, most previous studies measure the temperature at a >0.5 mm distance from the bone surface in contact with the drill, but the heat damages occur at the superficial tissues at the contact [5]. Despite recent efforts to measure the temperature directly from the drill, heat produced from the friction between the thermocouple and drill was ignored in the previous studies [14,15]. In this study, we have custom-built a drill equipped with a rotating thermocouple to eliminate the friction. We then measured the temperature from the inside of the bone being drilled via heat flow in real time. The temperature was attained under several conditions that vary in drilling speeds and depths. The resulting measurements were different among various drilling speeds and depths, which rejects the null hypothesis.

Considering the temperature changes based on bone depth, the heat was produced mostly at the drill tip where most friction and compressive forces are applied [20]. The infrared ray sensor method used in previous reports does not fully reflect the actual temperature changes during implant drilling because the temperature on the upper part of the drill did not increase in the same manner as the lower portion of the drill inside the bone during drilling, and the infrared ray sensor measured the temperature at the entrance of the drill hole only at the onset of drilling [21].

In this study, an 800 g weight was used according to previous drilling studies (when an oral maxillofacial surgeon drilled bone, a 6–24 N pressure was applied) [22,23]. Implantation torque was set at 30 N cm as this was shown to be the minimum torque that did not place excessive pressure on the bone around the implant but fixed it sufficiently in place [24,25]. The drill rotating speed recommended by manufacturers differs from 1000 to 1500 rpm, and we used from 500 to 2000 rpm [21]. As we increased the rotating speed, friction force and minimal temperature also increased, but the drilling time needed for reaching the intended depth decreased [26]. In this study, at higher rotation speed, the temperature increased more rapidly and the maximum temperature was also higher as shown in previous studies, indicating that the temperature of bone increased when the drill rotating

speed increased [27]. The patterns of temperature change observed herein were consistent at all three rotating speeds.

A stainless steel drill, frequently used for dental drilling, has a 14-fold greater thermal conductivity and a 30-fold greater thermal diffusion disparity compared to compact bones [13,28]. Evaluating the heat with a thermocouple located in bone where the distance is less than 0.5 mm was impossible because the drilling destroyed the thin bone layer between the thermocouple and the drill. Here, we evaluated the temperature of the drill using a rotating thermocouple in direct contact. Importantly, we observed that the heat measured using the previous methods that measured the direct drilling temperature without the rotating slip ring on a thermosensor was much higher than our methods equipped with the rotating slip ring on a thermocouple [14]. This higher drill temperature is attributed to the frictional heat generated between the stationary thermocouple used in the previous works.

Osseous tissues were deformed when the temperature was higher than 56 °C, and immediate damage of osseous tissues were observed at >53 °C in one second [29–31]. In this study, we observed a clear disparity in the maximum temperature and temperature changes of the drill compared to those of the bone. The temperature at the drill–bone interface exceeded 80 °C when the drill was used according to the manufacturer's instructions; this can result in potentially significant bone damage [29]. In addition, we discovered that the actual temperature on the drill is much higher than the temperature measured using previously reported methods [18]. The temperature of the drill exceeded 50, 60, and 80 °C, respectively at 500, 1200, and 2000 rpm drilling speed, much earlier than the thermocouple placed inside the bone, which reached its highest temperature of ~40, 45, and 50 °C, respectively at 500, 1200, and 2000 rpm drilling speed due to slow heat flow in the bone.

The thermocouple located at a bone depth of 10 mm measured the highest temperature. This may be due to more friction applied at the top of the drill tips and minimal heat loss due to increased depth. Previous studies revealed that external irrigation does not provide sufficient bone cooling as the drilling depth increases [20,30]. As human blood circulation within the bone allows some degree of cooling during operations, in vitro experiments face limits regenerating this cooling system using artificial bone models or cadavers [32]. Hence, the actual temperature in clinical situations might be lower and safer to the bone tissues [1]. Nevertheless, considering that the osseous tissues are damaged above 53 °C in one second of exposure, the result that the temperature exceeded ~80 °C after 5 s of drilling should demand serious attention by clinicians to avoid the risk of failure when drilling for dental implant [29–31].

## 5. Conclusions

In a preclinical implant drilling study, the actual temperature measured at the drill surface was significantly higher than the temperature measured inside the bone reported in previous studies. Despite the short duration of drilling, this high heat generated by the friction between drill and bone can exert a negative influence on the native bone tissues in osteotomy. This study provides a more clinically relevant mechanical study set up for the evaluation of the implant drilling, and can be a blueprint to design a more plausible implant drilling protocol to prevent bone tissue damage in various clinical conditions.

**Author Contributions:** Conceptualization, Y.K.; methodology, M.P.; software, S.J. (Sungwon Ju); validation, S.J. (Sungwon Ju); formal analysis, S.J. (Sungwon Ju); investigation, M.K.; resources, Y.K.; data curation, M.P.; writing—original draft preparation, Y.K.; writing—review and editing, M.K.; visualization, M.P.; supervision, S.J. (Sangho Jun) and J.A.; project administration, J.A.; funding acquisition, J.A. Both Y.K. and S.J. (Sungwon Ju) have equally contributed to this work. Both S.J. (Sangho Jun) and J.A. have equally contributed to this work.

**Funding:** This research was funded through Marine Biotechnology program by the Ministry of Oceans and Fisheries, grant number D11013214H480000110.

**Conflicts of Interest:** The authors declare no conflict of interest.

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
