# Peer review of "Direct Measurement of Heat Produced during Drilling for Implant Site Preparation"

_applsci, doi:10.3390/app9091898_

Round 1
Reviewer 1 Report
In the manuscript entitled: “Direct Measurement of Heat Produced During Drilling for Implant Site Preparation” the authors measured the temperature of the drill during implant site preparation.
In their study the temperature change of the drill was measured using a thermocouple inserted inside the stainless steel drill using an 18-gauge needle and mercury-containing slip ring. The drill rotating speed was varied from 500 rpm to 2000 rpm. Each drilling procedure was performed 10 times, and the average was calculated.
The authors found that the temperature of the drill increased rapidly, and the thermocouples in the bone reached to maximum temperature after the drill temperature started to decrease. The maximum inner temperature of the bone was the highest at a depth of 10 mm.
The authors concluded that the actual maximum temperature at drill and bone interface exhibited significantly higher than the temperature measured inside bone in the previous reports.
Major comments:
In general, the idea and innovation of this study, regards the analysis of direct measurement of heat produced during drilling for implant site preparation is interesting, because the role of the factors of xenogenic bone is validated but further studies on this topic could be an innovative issue in this field could be open an innovative matter of debate in literature by adding new information. Moreover, there are few reports in the literature that studied this interesting topic with this kind of study design.
The study was well conducted by the authors; However, there are some concerns to revise that are described below.
The introduction section resumes the existing knowledge regarding the important factor linked with factors which affect bone around implant.
However, as the importance of the topic, the reviewer strongly recommends, before a further re-evaluation of the manuscript, to update the literature through read, discuss and cites in the references with great attention all of those recent interesting articles, that helps the authors to better introduce and discuss the aim of the study in light of some other conditions linked with alteration in bone augmentation and adjuvants that enhance tissue healing: Isola G, Matarese M, Ramaglia L, Iorio-Siciliano V, Cordasco G, Matarese G. Efficacy of a drug composed of herbal extracts on postoperative discomfort after surgical removal of impacted mandibular third molar: a randomized, triple-blind, controlled clinical trial. Clin Oral Investig. 2018 Oct 11. doi: 10.1007/s00784-018-2690-9.
The authors should be better specified, at the end of the introduction section, the rational of the study and the aim of the study with the null hypothesis. In the material and methods section, should better clarify how was performed the equipment set up. Moreover, specify if was performed, the intra-examiner agreement for the analyses and if the data obtained were normalized or not. Please specify if was used a test unit.
The discussion section appears well organized with the relevant paper that support the conclusions, even if the authors should better discuss the relationship between periodontitis and osseointegration loss. The conclusion should reinforce in light of the discussions.
In conclusion, I am sure that the authors are fine clinicians who achieve very nice results with their adopted protocol. However, this study, in my view does not in its current form satisfy a very high scientific requirement for publication in this journal and requests a revision before publication.
Minor Comments:
Abstract:
- Better formulate the introduction section by better describing the aim of the study
Introduction:
- Please refer to major comments
Discussion
- Please add a specific sentence that clarifies the results obtained in the first part of the discussion
- Page 7 last paragraph: Please reorganize this paragraph that is not clear
Author Response
Point 1: Update the literature through read, discuss and cites in the references with great attention all of those recent interesting articles, that helps the authors to better introduce and discuss the aim of the study in light of some other conditions linked with alteration in bone augmentation and adjuvants that enhance tissue healing.
Response 1: We have added your suggested study and another recent study to our reference. Thank you.
Point 2: The authors should be better specified, at the end of the introduction section, the rational of the study and the aim of the study with the null hypothesis.
Response 2: We have revised the sentence so it would be more specific and added a description of our null hypothesis. This null hypothesis is later rejected in the discussion section. Thank you.
Point 3: Please add a specific sentence that clarifies the results obtained in the first part of the discussion.
Response 3: We have revised the paragraph so it provides clear explanation of our results. Thank you.
Point 4: Page 7 last paragraph: Please reorganize this paragraph that is not clear
Response 4: We have revised the paragraph to clarify confusing remarks. Thank you.
Thank you very much for your time and advice.

Reviewer 2 Report
This article will give us important and valuable information about relationship between drilling speed and temprature around drilled bone. I would like to express my respect to the authors. And this article would be suitable for applied sciences. However, for accept and publish this article, some consideration and modification would be needed.
In materials and methods authors use 1200 rpm for drill roating speed. However, recently, most implant campany recommend that drill roating speed should be 400~800 rpm. And even if external irrigation did not provide sufficient bone cooling from previous report, we usually use external irrigation in clinical situation. Authors should comment about those estrangemant from clinical condition and experimental setting in discussion.
Author Response
Point 1: In materials and methods authors use 1200 rpm for drill rotating speed. However, recently, most implant company recommend that drill rotating speed should be 400~800 rpm.
Response 1: We have updated our paragraph to provide explanation to your question. The 3 different rpms were chosen according to the manufacturers’ recommendation and possible clinical situations when the clinician perform drilling at slower and faster rpms. Thank you.
Point 2: And even if external irrigation did not provide sufficient bone cooling from previous report, we usually use external irrigation in clinical situation.
Response 2: We have added explanation to your question. Since infrared sensor detects water that is above the bone surface, we have decided to experiment under dry condition. Thank you.
Thank you very much for your time and advice.

Round 2
Reviewer 1 Report
In the R1 version of the manuscript entitled: “Direct Measurement of Heat Produced During Drilling for Implant Site Preparation” the authors followed all the issues suggested by the reviewer. Though the changes based on the reviewer comments, almost of the criticisms were carefully analysed and solved.
I have carefully evaluated all parts of the manuscript. I believe that the article, in this version, is now adequate for publication in this journal.